# Study of Nonlinear Aerodynamic Self-Excited Force in Flutter Bifurcation and Limit Cycle Oscillation of Long-Span Suspension Bridge

**Jieshan Liu** [1,2,3], **Fan Wang** [1,2,*] **and Yang Yang** [1,2]

1   School of Mechanics and Construction Engineering, Jinan University, Guangzhou 510632, China; kevinjason2010@hotmail.com (J.L.); yangyangly2012@gmail.com (Y.Y.)
2   Key Laboratory of Disaster Forecast and Control in Engineering, Ministry of Education, Guangzhou 510632, China
3   CCCC Fourth Harbor Engineering Institute Co., Ltd., Guangzhou 510230, China
*   Correspondence: twfan@jnu.edu.cn

**Abstract:** This article establishes a nonlinear flutter system for a long-span suspension bridge, aiming to analyze its supercritical flutter response under the influence of nonlinear aerodynamic self-excited force. By fitting the experimental discrete values of flutter derivatives using the least squares method, a polynomial function of flutter derivatives with respect to reduced wind speed is obtained. Flutter critical value is determined by the linear matrix eigenvalues of a state-space equation. The occurrence of a supercritical Hopf bifurcation in the nonlinear system is determined by the Jacobian matrix eigenvalues of the state-space equation and the system's vibrational response at the critical state. The vibrational response of the supercritical state is obtained through Runge–Kutta integration, revealing the presence of stable limit cycle oscillation (LCO) and unstable limit cycle oscillation in the system, and through analyzing the relationship between the LCO amplitude and wind speed. Considering cubic nonlinear damping and stiffness, the effects of different factors on the nonlinear flutter system are analyzed.

**Keywords:** flutter bifurcation; nonlinear aerodynamic self-excited force; nonlinear damping; nonlinear stiffness; supercritical Hopf bifurcation; limit cycle oscillation (LCO)

## 1. Introduction

Since the issue of bridge flutter was put forward, numerous studies have been conducted on aerodynamic vibrations in bridges. Flutter, as a form of aerodynamic instability, is the most destructive type of vibration induced by wind. When wind speed is low, structural vibrations caused by the interaction between the structure and the flow field are dissipated by system damping, keeping the bridge in a stationary state. However, when wind speed exceeds a critical value, the motion energy generated by the interaction between the structure and the flow field cannot be completely dissipated by system damping, resulting in flutter. Flutter is a self-excited vibration phenomenon of elastic structures induced by their interaction with fluid in the flow field. This study investigates the nonlinear flutter response when wind speed exceeds the critical value. The occurrence of self-limiting flutter, known as limit cycle oscillations in bridges, has been indicated by the Tacoma Narrows Bridge flutter event and recent research and experiments. This type of limit cycle oscillation, often referred to as "hard flutter," differs from linear divergence-type flutter theories and does not always lead to divergence [1–5]. Notably, nonlinearity in the aerodynamic self-excited forces of bluff body bridges is significant [6–9], and the nonlinear phenomenon stands out under angle of attack conditions [10,11]. Moreover, there exists a significant potential for advancing our assessment of the vibrational performance of bridges through machine learning algorithms based on artificial neural networks. These

algorithms employ geometric information and dynamic parameters as inputs to construct vibrational models of bridges. The integration of computational fluid dynamics (CFD) with machine learning models is used to predict wind loads, and it has demonstrated satisfactory results in forecasting and evaluating the aerodynamic elastic responses of bridges [12–14]. This underscores the widespread attention garnered by the study of nonlinear flutter in long-span bridges. Although bridge design still does not allow for situations beyond the flutter critical wind speed, studying flutter behavior beyond the critical state contributes to a deeper understanding of the structural responses of bridges under flutter conditions.

The flutter instability process caused by aeroelastic effects is generally analyzed using the aerodynamic self-excitation theory proposed by Scanlan et al. [15]. Traditional studies have established linear flutter equations based on this theory, but the limitations of linear systems are evident. They can only solve the flutter critical state of the aerodynamic self-excited system. For example, Náprstek et al. proposed stability conditions and unified linear variables for bluff bridge deck sections under unsteady flow [16,17]. However, as the span of suspension bridges increases, the vibration response of bridges after flutter becomes increasingly important. Therefore, it is crucial to establish nonlinear motion equations for self-excited flutter in suspension bridges.

In the conventional Scanlan aerodynamic self-excited force model, the interaction between the bridge cross-section and aerodynamic forces is represented by a linear multiplication of flutter derivatives and motion components. The higher-order terms of flutter derivatives are disregarded. However, the aerodynamic self-excited force is inherently nonlinear. Considering the critical post-flutter of bridges, it becomes essential to formulate the appropriate equations to describe this nonlinear motion. Numerous scholars have delved into the study of nonlinear flutter. Building upon Scanlan's research, they have demonstrated a strong correlation between the flutter derivatives related to torsional motion and nonlinearity. They have determined a functional relationship between nonlinear flutter derivatives, reduced wind speed, and flutter amplitudes [2,11]. Furthermore, through wind tunnel experiments with box girder models, they have proposed a nonlinear aeroelastic self-excited force model, coupling bridge lift and torsion [18]. Additionally, investigations into complex nonlinear fluid–structure interaction phenomena in turbulent flow around bluff bodies have been conducted through wind tunnel tests and numerical simulations [19–21]. It has been found that the presence of a strong shear layer flow separation contributes significantly to the nonlinearity of aerodynamic self-excited forces, manifested in variations in the amplitude of the first harmonic and the appearance of higher harmonics. As for the nonlinear motion system of bridge flutter, researchers have focused on the critical bifurcation state and post-bifurcation behavior. Research has mainly centered on analyzing various mechanical and aerodynamic characteristics (such as mechanical damping ratio, natural frequency, initial angle of attack, and flutter derivatives) and their influence on the motion behavior of a given cross-section after the critical flutter state [22–24]. Moreover, the critical state between static and flutter behavior of nonlinear systems is represented by Hopf bifurcations [25]. Measures to suppress the occurrence of flutter motion require attention, primarily in terms of optimizing and controlling the placement of tuned mass dampers [26,27]. However, previous studies have often concentrated on specific local aspects of nonlinear motion systems, such as establishing wind tunnel models corresponding to bridge structures to analyze flutter motion states, exploring the acquisition and identification of high-order flutter derivatives in nonlinear aerodynamic self-excited forces, and investigating effective energy dissipation and vibration suppression measures post flutter. There remains a relatively limited amount of research focused on establishing nonlinear motion equations for bridge flutter systems and studying critical flutter states and post-flutter motion behaviors.

Based on the status of current research, it is evident that nonlinear aerodynamic self-excitation exists, and the discrete values of the nonlinear components of flutter derivatives can be obtained through wind tunnel section model tests. Therefore, this study considers the effect of nonlinear aerodynamic self-excited force on bridges, establishes nonlinear

flutter equations for long-span suspension bridges, and fits the function expression of flutter derivatives with respect to reduced wind speed based on the discrete values of flutter derivatives obtained from the literature. Subsequently, the nonlinear bifurcation and limit cycle oscillation phenomena are investigated based on the motion equations, analyzing and determining the bifurcation types and studying the properties of LCO, as well as predicting the development of the relationship between LCO and wind speed. Finally, on this basis, the effects of nonlinear damping and structural nonlinear stiffness on flutter LCO are further considered.

## 2. Framework of Nonlinear Flutter Analysis

In this section, motion equations for a long-span suspension bridge under the influence of nonlinear aerodynamic self-excited force are established. Flutter equations are obtained by fitting the discrete values of flutter derivatives in the nonlinear aerodynamic force model using the least squares method. The framework also considers the nonlinear stiffness and nonlinear damping of the structure, and four computational analysis cases are set up.

### 2.1. Flutter Motion Equations for Suspension Bridges

The current research on flutter utilizes the motion expression proposed by Scanlan [28]. This model considers bridges as Euler–Bernoulli beams, thereby neglecting shear deformations. Furthermore, it assumes that the plane (cross-section), which is perpendicular to the beam's central axis before deformation, remains planar after deformation, and that regardless of pre- and post-deformation processes, the plane of the cross-section remains perpendicular to the beam's axis. Because the centroid of the bridge's cross-section coincides with its shear center, the vibration equation for the beam, derived based on the Euler–Bernoulli beam theory and the method of separation of variables, can be expressed in terms of its natural modes, mass, and stiffness. The first-mode response of the bridge contributes significantly to its vibration, resulting in a two-degree-of-freedom cross-sectional motion model consisting of a first-order vertical bending and torsional frequencies of the bridge, along with their corresponding masses and stiffness.

As shown in Figure 1, the cross-section of the bridge deck of the Nansha Bridge over the Nizhou Channel is allowed to undergo motion in the vertical bending and torsional degrees of freedom. It experiences aerodynamic self-excitation forces in a uniform flow field, including aerodynamic lift and aerodynamic torque, acting at the centroid of the cross-section [15,28]. Therefore, flutter motion equations can be expressed as follows. On the left side of the equation lies the motion equation representing the characteristics of the bridge, while on the right side lies the aeroelastic self-exciting force exerted on the bridge:

$$m_h\ddot{h} + 2\xi_h m_h\omega_h\dot{h} + m_h\omega_h^2 h = \rho U^2 B\left(KH_1^*\frac{\dot{h}}{U} + KH_2^*\frac{B\dot{\alpha}}{U} + K^2H_3^*\alpha + K^2H_4^*\frac{h}{B}\right) \quad (1)$$

$$I\ddot{\alpha} + 2\xi_\alpha I\omega_\alpha\dot{\alpha} + I\omega_\alpha^2\alpha = \rho U^2 B^2\left(KA_1^*\frac{\dot{h}}{U} + KA_2^*\frac{B\dot{\alpha}}{U} + K^2A_3^*\alpha + K^2A_4^*\frac{h}{B}\right) \quad (2)$$

where $m$ and $I$ are mass and moment of inertia per unit length of the bridge deck, respectively. $h$, $\dot{h}$, and $\ddot{h}$ represent the displacement, velocity, and acceleration of the vertical bending degree of freedom, while $\alpha$, $\dot{\alpha}$, and $\ddot{\alpha}$ represent the angular displacement, angular velocity, and angular acceleration of the torsional degree of freedom. $\xi_h$ and $\xi_\alpha$ are the damping ratios of the suspension bridge in the vertical bending and torsional modes, respectively. $\omega_h$ and $\omega_\alpha$ are the first-order circular frequencies of vertical bending and torsion, respectively. On the right-hand side of Equation (1) is the aerodynamic lift, and on the right-hand side of Equation (2) is the aerodynamic torque. $\rho$ is air density, $U$ is fluid flow velocity, $B$ is the width of the cross-section of the stiffened beam, and $K$ is reduced frequency, defined as $K = B\omega/U$. $H_i^*$ and $A_i^*$ are eight flutter derivatives, which are

dimensionless quantities related to the reduced frequency or reduced wind speed and can be expressed as functions.

$$H_i^*, A_i^* = f(V_r) = f(K) \tag{3}$$

where $V_r$ is the reduced wind speed, defined as $V_r = U/Bf$. This study focuses on the Nansha Bridge over the Nizhou Channel, which is a suspension bridge with a main span of 1688 m and a bridge deck width of 41.7 m. The relevant design parameters of the bridge are listed in Table 1 [29].

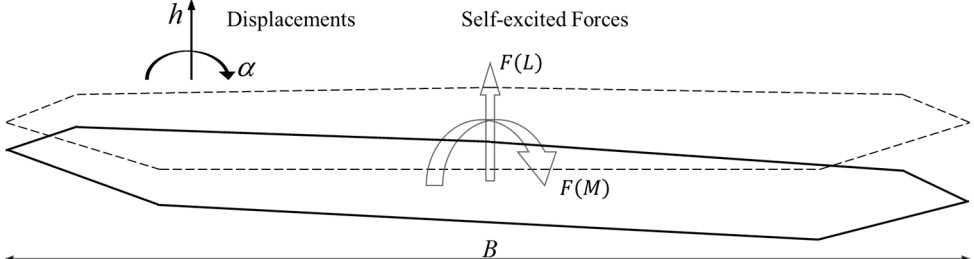

**Figure 1.** Motion model of bridge deck section.

**Table 1.** Parameter values of the suspension bridge.

| Parameter | Value |
|:---:|:---:|
| $m$ | 65,649 (kg/m) |
| $I$ | 7,808,150 (kg·m) |
| $B$ | 41.7 (m) |
| $\xi_h$ | 0.005 |
| $\xi_\alpha$ | 0.005 |
| $\omega_h$ | 0.45497 (rad/s) |
| $\omega_\alpha$ | 1.33285 (rad/s) |

### 2.2. Cubic Damping and Cubic Torsional Stiffness

Taking into account the structural nonlinearity of the bridge's stiffness and the possibility of obtaining strong nonlinear damping using tuned mass dampers, the vibration equations of the abovementioned flutter motion Equations (1) and (2) are expressed in the following form:

$$m_h \ddot{h} + C_h\left(\dot{h}\right) + k_h h = F(L) \tag{4}$$

$$I \ddot{\alpha} + C_\alpha\left(\dot{\alpha}\right) + k_\alpha(\alpha) = F(M) \tag{5}$$

where $C_h\left(\dot{h}\right)$ and $C_\alpha\left(\dot{\alpha}\right)$ represent the nonlinear damping values of the flutter system in the vertical bending and torsional degrees of freedom, $k_\alpha(\alpha)$ is structural nonlinear stiffness, and $F(L)$ and $F(M)$ are self-excited forces of flutter.

The strong nonlinear damping in the vertical bending and torsional degrees of freedom is expressed as a function of vertical velocity $\dot{h}$ and angular velocity $\dot{\alpha}$; then, the vertical and torsional damping value $c_h$ and $c_\alpha$ are written as follows:

$$C_h\left(\dot{h}\right) = c_{h1}\dot{h} + c_{h2}\dot{h}^3 \tag{6}$$

$$C_\alpha\left(\dot{\alpha}\right) = c_{\alpha1}\dot{\alpha} + c_{\alpha2}\dot{\alpha}^3 \tag{7}$$

In the equation, linear damping coefficients are given by $c_{h1} = 2\xi_h m_h \omega_h$ and $c_{\alpha1} = 2\xi_\alpha I \omega_\alpha$.

Considering the nonlinear stiffness in the torsional degree of freedom of the suspension bridge, the bending and torsional stiffness are expressed as follows:

$$k_h h = m_h \omega_h^2 h \tag{8}$$

$$k_\alpha(\alpha) = k_{\alpha 1}\alpha + k_{\alpha 2}\alpha^3 \tag{9}$$

In the equation, stiffness $k_h$ in the vertical degree of freedom is set as a linear function of vertical displacement $h$. $k_\alpha$, $k_{\alpha 1}$ represent the first-order coefficient of torsional stiffness, while $k_{\alpha 1} = I\omega_\alpha^2$, and $k_{\alpha 2}$ represent the cubic term coefficient of torsional stiffness.

The cubic damping coefficient and cubic torsional coefficient are determined in Table 2.

**Table 2.** Values of cubic damping and stiffness terms for each case.

| Cases Name | $c_{h2}$ | $c_{\alpha 2}$ | $k_{\alpha 2}$ |
|---|---|---|---|
| Reference case | 0 | 0 | 0 |
| Case A | 0 | $10c_{\alpha 1}$ | 0 |
| Case B | 0 | $10c_{\alpha 1}$ | $1000m_h\omega_\alpha^2$ |
| Case C | $10c_{h1}$ | $10c_{\alpha 1}$ | $1000m_h\omega_\alpha^2$ |

*2.3. Expression of Nonlinear Aerodynamic Self-Excited Force*

Based on the aerodynamic self-excited force model proposed by Scanlan (Equations (1) and (2)), Gao et al. proposed a nonlinear aerodynamic self-excited force model [18]. The expressions for the nonlinear self-excited forces $F(L)$ and $F(M)$ in this model are as follows:

$$F(L) = \rho U^2 B\left[KH_1^*\frac{\dot{h}}{U} + \left(KH_2^* + KH_{2,02}^*\frac{B|\dot{\alpha}|}{U}\right)\frac{B\dot{\alpha}}{U} + (K^2H_3^* + K^2H_{3,02}^*|\alpha|)\alpha + K^2H_4^*\frac{h}{B} + K^2H_{4,01}^*|\alpha| + K^2H_{4,02}^*\alpha^2\right] \tag{10}$$

$$F(M) = \rho U^2 B^2\left[KA_1^*\frac{\dot{h}}{U} + \left(KA_2^* + KA_{2,02}^*\frac{B|\dot{\alpha}|}{U} + KA_{2,03}^*\frac{B^2\dot{\alpha}^2}{U^2}\right)\frac{B\dot{\alpha}}{U} + K^2A_3^*\alpha + K^2A_4^*\frac{h}{B}\right] \tag{11}$$

where $F_L$ represents aerodynamic lift force, $F_M$ represents aerodynamic torque, $\rho$ is air density, $U$ is fluid velocity, $B$ is the cross-sectional width of the girder, $K$ is the reduced frequency, $K = B\omega/U$. $H_i^*$ and $A_i^*$ are flutter derivatives, and $H_{2,02}^*$, $H_{3,02}^*$, $H_{4,01}^*$, $H_{4,02}^*$, $A_{2,02}^*$, $A_{2,03}^*$ are small perturbation flutter derivatives. These flutter derivatives are dimensionless quantities that depend on the reduced frequency or reduced wind speed. Similarly, these flutter derivatives can be expressed as functions of the reduced wind speed or reduced frequency. For computational convenience, in this study, these flutter derivatives are expressed as functions of the reduced wind speed.

$$H_i^*, A_i^* = f(V_r) = f(K) = H, A_{i1}^* \cdot V_r + H, A_{i2}^* \cdot V_r^2 \tag{12}$$

$$H_2^* \vee A_2^* = f(V_r) = f(K) = H_{i1}^* \vee A_{i1}^* \cdot V_r + H_{i2}^* \vee A_{i2}^* \cdot V_r^2 + H_{i3}^* \vee A_{i3}^* \cdot V_r^2 + H_{i4}^* \vee A_{i4}^* \cdot V_r^2 \tag{13}$$

Nonlinear flutter self-excited forces (10) and (11) not only include linear first-order components but also nonlinear second-order components. In the nonlinear self-excited force model proposed by Gao et al., the discrete results of flutter derivatives were obtained through wind tunnel tests on a bridge deck section model [18]. In the flutter motion equation proposed by Scanlan, the flutter derivatives in the flutter self-excited forces are only related to the cross-sectional shape of the bridge deck panel. Therefore, it can be assumed that the numerically obtained results of the box girder section model by Gao et al. can be used to analyze the nonlinear self-excited forces of the Nansha Bridge over the Nizhou Channel in this study. To facilitate a continuous solution of the flutter motion

equation, the flutter derivatives are fitted as a function of the reduced wind speed using the least squares method, as shown in the form of (12). The discrete values of the first-order flutter derivatives and the corresponding function graph are shown in Figure 2, where the black dots represent the discrete flutter derivative values obtained from wind tunnel tests, and the black solid line represents the fitted flutter derivative function curve obtained through the least squares method.

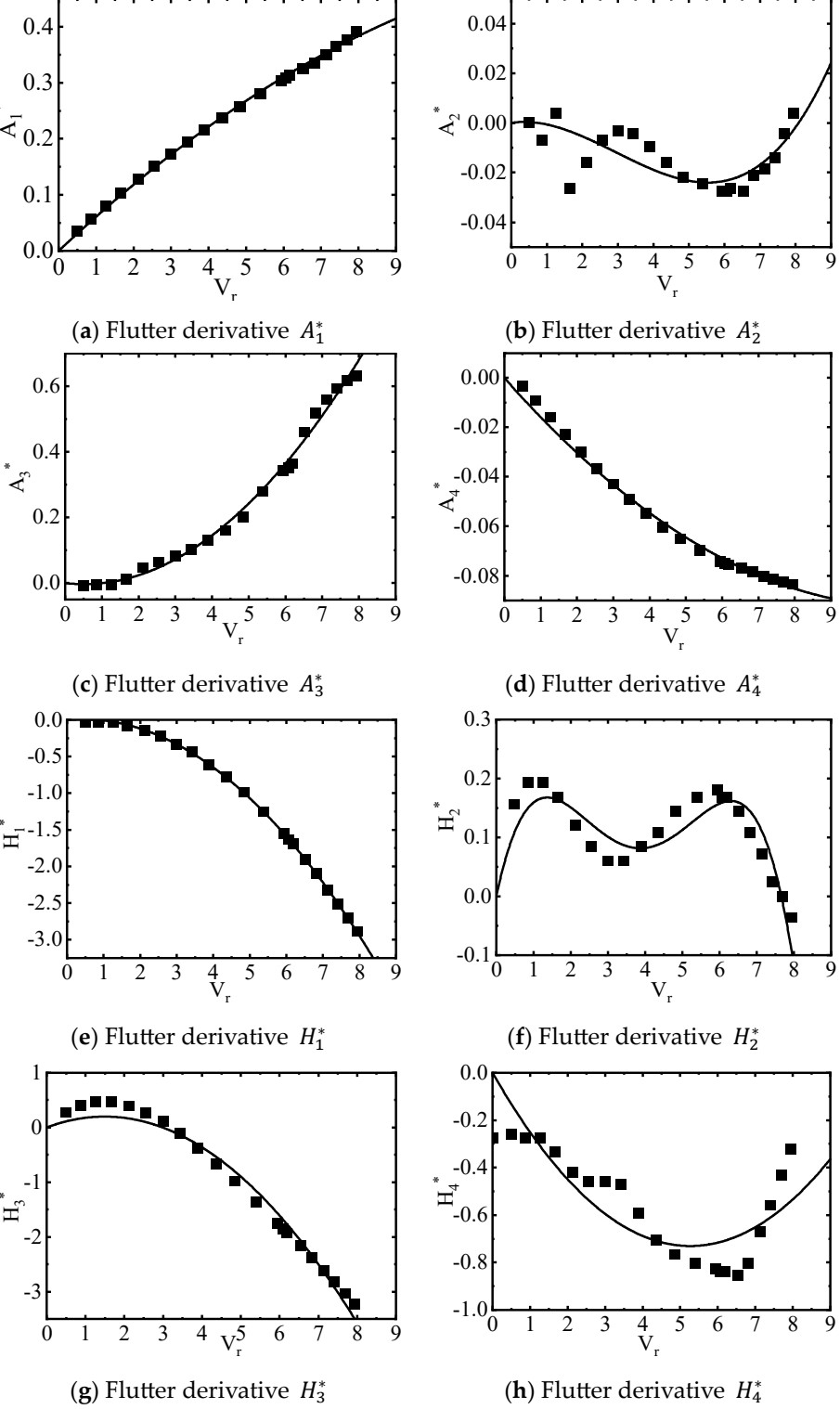

**Figure 2.** Discrete points and fitted curves of the first-order flutter derivatives of aero self-excited force.

The discrete values of the second-order flutter derivatives are shown in Figure 3, where the red color represents the second-order flutter derivatives $H_{2,02}^*$, $H_{3,02}^*$, $H_{4,01}^*$, $H_{4,02}^*$, $A_{2,02}^*$, $A_{2,03}^*$. The red dots represent the discrete values obtained from wind tunnel tests, and the red solid line represents the continuous function curve obtained through fitting.

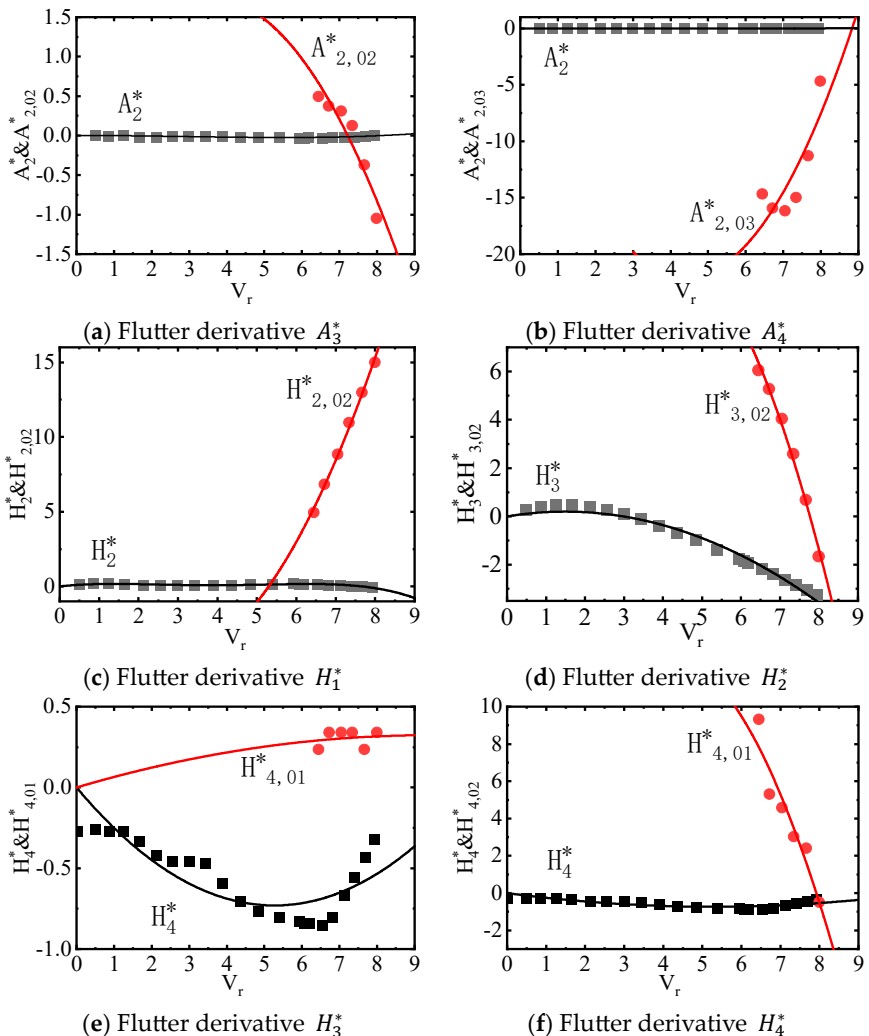

**Figure 3.** Discrete points and fitted curves (colored in red) of the second-order flutter derivatives of aero self-excited force.

The coefficients of the function expressions for the continuous curves of flutter derivatives shown in Figures 2 and 3 are listed in Tables 3 and 4.

**Table 3.** The coefficients in the curve of the fitting quadratic function of the flutter derivative.

| | $H_{i1}^*$ | $H_{i2}^*$ | | $A_{i1}^*$ | $A_{i2}^*$ |
|---|---|---|---|---|---|
| $H_1^*$ | 0.04374 | −0.05154 | $A_1^*$ | 0.06288 | −0.00187 |
| $H_3^*$ | 0.26574 | −0.08902 | $A_3^*$ | −0.01230 | 0.01217 |
| $H_4^*$ | −0.27729 | 0.02632 | $A_4^*$ | −0.01665 | $7.49839 \times 10^{-4}$ |
| $H_{2,02}^*$ | −3.76111 | 0.70966 | $A_{2,02}^*$ | 0.95929 | −0.13286 |
| $H_{3,02}^*$ | 5.81372 | −0.74839 | $A_{2,03}^*$ | −9.99665 | 1.13183 |
| $H_{4,01}^*$ | 0.06882 | −0.00366 | | | |
| $H_{4,02}^*$ | 6.52151 | −0.82327 | | | |

**Table 4.** The coefficients in the curve of the fitting quartic function of the flutter derivative.

| | $H_{i1}^* \vee A_{i1}^*$ | $H_{i2}^* \vee A_{i2}^*$ | $H_{i3}^* \vee A_{i3}^*$ | $H_{i4}^* \vee A_{i4}^*$ |
|---|---|---|---|---|
| entry 1 | 0.29832 | −0.17132 | 0.03441 | −0.00224 |
| entry 2 | 0.00214 | −0.00314 | $3.55170 \times 10^{-4}$ | 0 |

## 3. Flutter Critical State and Hopf Bifurcation

According to the findings of Andronov et al., if a linear expression of a system's motion considers the critical point as a saddle point, a node, or a focus of the critical state, then for the original nonlinear system, the critical point is also a saddle point, a node, or a focus [30]. The determination of the critical point properties of a nonlinear system can be made by examining the Jacobian matrix of the motion equations and the motion states in the vicinity of the critical values. In this section, using the values of the reference case in Table 2, the calculation methods for the critical state, bifurcation determination, and bifurcation limit cycle of the nonlinear self-excited flutter system of the suspension bridge are presented.

### 3.1. Flutter Critical Wind Speed Solution

Flutter Equations (4) and (5) can be written in the form of state-space equations, i.e., $\dot{x} = f(x)$. Let $x = \left[h, \alpha, \dot{h}, \dot{\alpha}\right]^T$; thus, the state-space equations of this nonlinear flutter system can be written in the following form:

$$\dot{x} = F(U, \omega, x) = F_{Linear} \cdot x + F_{Nonlinear}(x) \tag{14}$$

where $F_{Linear}$ is the linear coefficient matrix of the flutter equations, and $F_{Nonlinear}$ is the nonlinear term of the flutter equations. Based on the analysis and calculations of the nonlinear system equations in the previous sections, the critical flutter state can be computed using the linear part of Equation (14), and then the complete state-space equations can be used to analyze the nonlinear flutter response. The coefficients of the linear and nonlinear parts of Equation (14) are given as follows:

$$F_{Linear} = \rho U^2 B \begin{bmatrix} 0 & 0 & 1 & 0 \\ 0 & 0 & 0 & 1 \\ \frac{K^2 H_4^*}{B m_h} - \frac{k_h}{m_h} & \frac{K^2 H_3^*}{m_h} & \frac{K H_1^*}{U m_h} - \frac{c_h}{m_h} & \frac{K H_2^* B}{U m_h} \\ \frac{K^2 A_4^*}{I} & \frac{B K^2 A_3^*}{I} - \frac{k_\alpha}{I} & \frac{B K A_1^*}{U I} & \frac{B^2 K A_2^*}{U I} - \frac{c_\alpha}{I} \end{bmatrix} \tag{15}$$

$$F_{Nonlinear}(x) = \begin{bmatrix} 0 \\ 0 \\ \frac{\rho U^2 B}{m_h}\left( K H_{2,02}^* \frac{B^2 |\dot{\alpha}| \dot{\alpha}}{U^2} + K^2 H_{3,02}^* |\alpha|\alpha + K^2 H_{4,01}^* |\alpha| + K^2 H_{4,02}^* \alpha^2 \right) \\ \frac{\rho U^2 B^2}{I}\left( K A_{2,02}^* \frac{B|\dot{\alpha}|}{U} + K A_{2,03}^* \frac{B^2 \dot{\alpha}^2}{U^2} \right)\frac{B\dot{\alpha}}{U} \end{bmatrix} \tag{16}$$

The critical point of a nonlinear motion system can be obtained by solving its linear expression. According to the Theodorsen method, as the wind speed $V_r$ of the system parameters changes, when the eigenvalues of the matrix (15) have positive real parts, the corresponding wind speed $V_{Cr}$ is the critical value of the system [31]. Therefore, the critical flutter state can be obtained through the following steps:

1.  Assume a small value of frequency $\omega_0$ and substitute it into the matrix (15). Gradually increase the reduced wind speed $V_r$ until the first pair of complex conjugate eigenvalues of the matrix have zero real parts. Record the imaginary part $Imag(\lambda)$ of this eigenvalue as frequency $\omega_i = Imag(\lambda)$;
2.  Take $\omega_i$ as the new frequency value and substitute it into the matrix (15). Again, gradually increase the reduced wind speed $V_r$ until the first complex eigenvalue of

the matrix has zero real parts. Record the imaginary part $Imag_i(\lambda)$ of this eigenvalue as the new flutter frequency $\omega_{i+1} = Imag_i(\lambda)$.;

3. Compare $\omega_i$ and $\omega_{i+1}$. Repeat step 2 until $|\omega_{i+1} - \omega_i|$ approaches zero. At this point, the frequency value is the critical flutter frequency $\omega_C$ of the system, the reduced wind speed value is the critical reduced wind speed $V_{Cr}$ of the flutter, and the flutter critical wind speed $U_C$ can be obtained using the relationship $V_r = 2U\pi/B\omega$.

Through the above steps, the parameter $V_r$ that causes the eigenvalues of the matrix $F_{Linear}$ to change from negative to positive is obtained, and the flutter system reaches the critical state. At this critical state, the matrix eigenvalues are given by:

$$\lambda|_{V_r=7.2673} = \begin{pmatrix} -0.0593 + 0.4754i \\ -0.0593 - 0.4754i \\ 0.0000 + 1.1888i \\ 0.0000 - 1.1888i \end{pmatrix}$$

The flutter system's critical flutter frequency is $\omega_C = 1.1888$ rad/s, the critical reduced wind speed is $V_{Cr} = 7.2673$, and the critical flutter wind speed is $U_C = 57.3374$ m/s.

*3.2. Proof of Hopf Bifurcation*

The Jacobian matrix of the state-space Equation (14) is obtained according to the rule in Equation (17).

$$J_{F(x)} = \begin{bmatrix} 0 & 0 & 1 & 0 \\ 0 & 0 & 0 & 1 \\ \frac{\partial \ddot{h}}{\partial h} & \frac{\partial \ddot{h}}{\partial \alpha} & \frac{\partial \ddot{h}}{\partial \dot{h}} & \frac{\partial \ddot{h}}{\partial \dot{\alpha}} \\ \frac{\partial \ddot{\alpha}}{\partial h} & \frac{\partial \ddot{\alpha}}{\partial \alpha} & \frac{\partial \ddot{\alpha}}{\partial \dot{h}} & \frac{\partial \ddot{\alpha}}{\partial \dot{\alpha}} \end{bmatrix} \tag{17}$$

Based on the results of the critical values of the flutter, taking the reduced wind speeds $V_r = 7$, $V_r = V_{cr} = 7.2673$, and $V_r = 7.5$, they are substituted into the Jacobian matrix to calculate the corresponding eigenvalues, as shown below:

$$\lambda|_{V_r=7} = \begin{pmatrix} -0.0543 + 0.4782i \\ -0.0543 - 0.4782i \\ -0.0022 + 1.2001i \\ -0.0022 - 1.2001i \end{pmatrix}$$

$$\lambda|_{V_r=7.2673} = \begin{pmatrix} -0.0593 + 0.4754i \\ -0.0593 - 0.4754i \\ 0.0000 + 1.1888i \\ 0.0000 - 1.1888i \end{pmatrix}$$

$$\lambda|_{V_r=7.5} = \begin{pmatrix} -0.0638 + 0.4725i \\ -0.0638 - 0.4725i \\ 0.0023 + 1.1785i \\ 0.0023 - 1.1785i \end{pmatrix}$$

The obtained eigenvalues from the three sets of results above indicate that the Jacobian matrix consistently has pairs of complex conjugate eigenvalues. Moreover, as the system parameter $V_r$ surpasses the flutter critical value from small to large, the Jacobian matrix exhibits a pair of complex conjugate eigenvalues with a change in the real part from negative to positive. This characteristic satisfies the definition of Hopf bifurcation. Therefore, the aforementioned process demonstrates that the nonlinear flutter system experiences Hopf bifurcation at the flutter critical state.

Furthermore, Hopf bifurcation encompasses three types of critical bifurcations. Next, the nonlinear flutter response in the intervals on both sides of the Hopf bifurcation point will be analyzed through calculations to further prove the Hopf bifurcation type. The

Runge–Kutta methods will be used to compute the motion response of this nonlinear flutter equation [32]. The flutter equations, as established in this paper, have been formulated into a first-order ordinary differential equation represented as Equation (14). Consequently, for the purpose of solving this equation, the adoption of the fourth-order Runge–Kutta method is appropriate. By carefully selecting an appropriate time step, the temporal response of the flutter can be obtained, thereby facilitating a more in-depth analysis of nonlinear vibrations.

- Wind speed: $U = 75$ m/s (beyond the critical state)

Initial state of motion: $\left(h, \dot{h}, \alpha, \dot{\alpha}\right)_1^T = (0, 0, 0.001, 0)^T$ (small perturbation).

Taking wind speed $U = 75$ m/s, setting the system vibration frequency $\omega = 1.888$ rad/s, and using the initial conditions for numerical integration $\left(h, \dot{h}, \alpha, \dot{\alpha}\right)_1^T = (0, 0, 0.001, 0)^T$, the vibration time history response is obtained as shown below. Additionally, the vibration limit cycle is plotted.

From Figures 4 and 5, it can be observed that when the wind speed $U = 75$ $m/s$, the vibration response of the nonlinear flutter system begins to diverge. This indicates that at this point, the focus of the flutter system, i.e., the zero point, becomes unstable. It becomes an unstable focus, and flutter occurs under the influence of a small perturbation $\alpha$=0.001 rad. The motion trajectory gradually moves away from this unstable focus. After a certain period of time, the flutter tends towards a stable state known as limit cycle oscillation. Figure 6 represents the phase portraits of Figures 4 and 5 in the velocity and displacement planes, illustrating the occurrence of stable limit cycle oscillation in the system at the current wind speed.

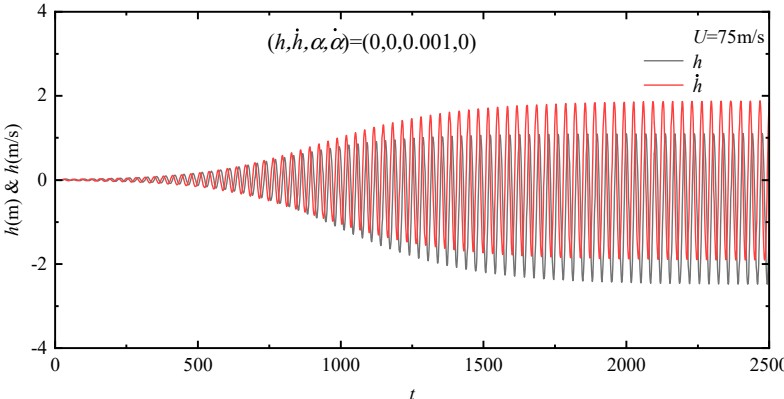

**Figure 4.** Continuous vibration of bridge flutter in vertical bending DOF at wind speed of 75 m/s and initial state of motion $\left(h, \dot{h}, \alpha, \dot{\alpha}\right)_1^T = (0, 0, 0.001, 0)^T$.

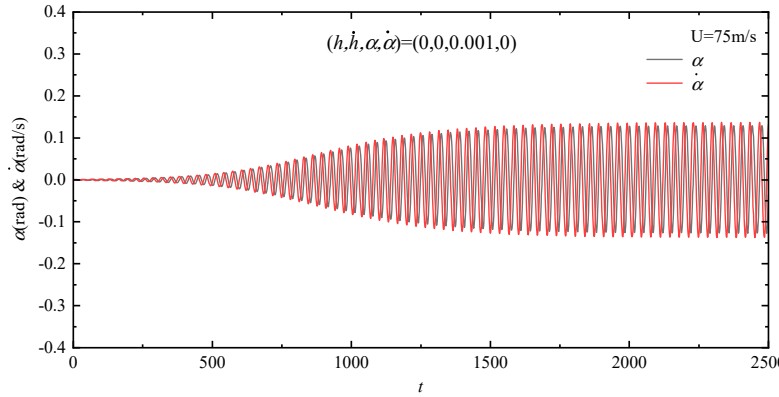

**Figure 5.** Continuous vibration of bridge flutter in torsional DOF at wind speed of 75 m/s and initial state of motion $\left(h, \dot{h}, \alpha, \dot{\alpha}\right)_1^T = (0, 0, 0.001, 0)^T$.

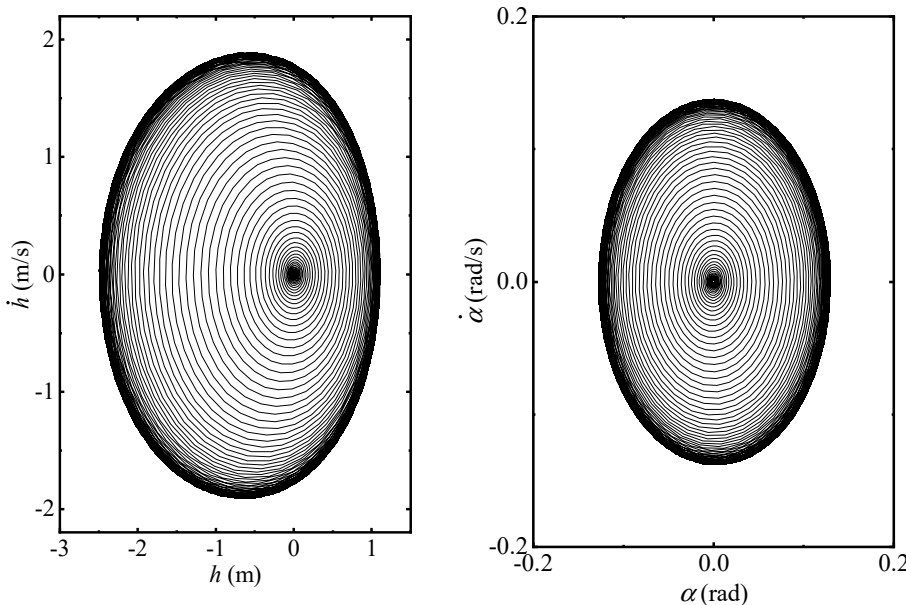

(**a**) Vibration LCO in vertical DOF  (**b**) Vibration LCO in torsional DOF

**Figure 6.** Phase trajectory diagram of LCO in vertical and torsional DOF at wind speed of 75 m/s and initial state of motion $\left(h, \dot{h}, \alpha, \dot{\alpha}\right)_1^T = (0, 0, 0.001, 0)^T$.

- Wind Speed: Wind speed $U = 75$ m/s (beyond the critical state),

  Initial state of motion: $\left(h, \dot{h}, \alpha, \dot{\alpha}\right)_2^T = (0, 0, 0.2, 0)^T$.

  To demonstrate that the LCO vibration is stable and repetitive, it is necessary to select initial conditions with amplitudes greater than the stable amplitude. Therefore, when the wind speed $U = 75$ m/s, the initial condition for the motion integration is set as $\left(h, \dot{h}, \alpha, \dot{\alpha}\right)_2^T = (0, 0, 0.2, 0)^T$. The system's time response is computed and shown below.

  From Figures 7 and 8, it can be observed that at the same wind speed $U = 75$ m/s, by only changing the torsional displacement $\alpha$ in the integration initial conditions to a value greater than the torsional amplitude of the limit cycle oscillation, the system's vibration response exhibits damping and eventually settles into a stable state after a certain period of time. Additionally, it can be noticed that the amplitude of the limit cycle oscillation, once it reaches the stable state, is the same as the stable limit cycle oscillation amplitudes shown in Figures 4 and 5. The stable limit cycle oscillation observed in the steady-state vibration response, after a period of time, is indeed a stable limit cycle. The phase plots of the two degrees of freedom are illustrated in Figure 9, with the motion trajectory being clockwise. The phase trajectory of the vertical bending degree of freedom starts at the origin, and the trajectory curve indicates a gradual convergence towards a closed circular loop. The phase trajectory of the torsional degree of freedom starts at $(\alpha, \dot{\alpha}) = (0.2, 0)$; similarly, the trajectory curve gradually approaches a closed circular loop. In comparison to the phase plot shown in Figure 6, the closed circular loop that the limit cycle oscillation trajectory converges towards is the same, proving that this closed circular loop is a stable limit cycle exhibited by the nonlinear vibration system at a wind speed of $U = 75$ m/s.

  From the aforementioned vibration of the limit cycle oscillation, it is evident that when wind speed exceeds the critical state, the system exhibits stable limit cycle oscillations, thereby demonstrating that the nonlinear vibration system undergoes a supercritical Hopf bifurcation at the critical state.

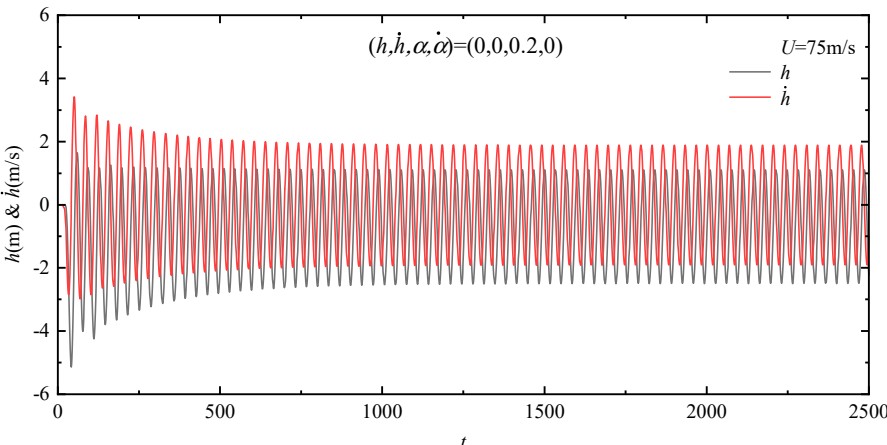

**Figure 7.** Continuous vibration of bridge flutter in vertical bending DOF at wind speed of 75 m/s and initial state of motion $\left(h, \dot{h}, \alpha, \dot{\alpha}\right)_2^T = (0, 0, 0.2, 0)^T$.

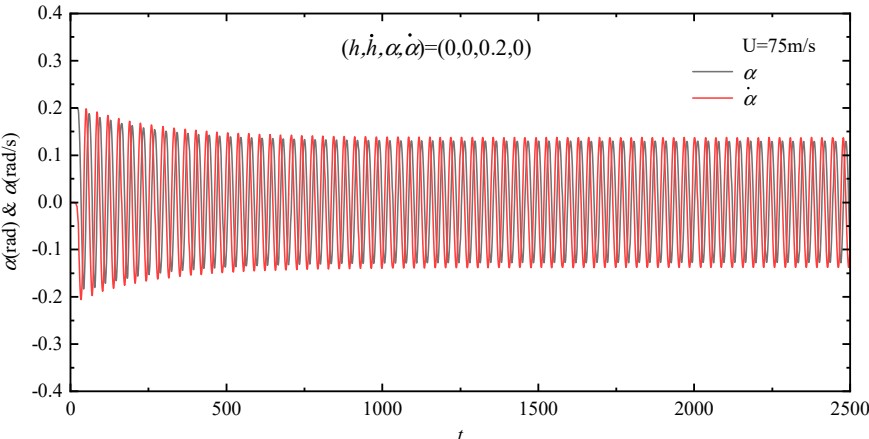

**Figure 8.** Continuous vibration of bridge flutter in torsional DOF at wind speed of 75 m/s and initial state of motion $\left(h, \dot{h}, \alpha, \dot{\alpha}\right)_2^T = (0, 0, 0.2, 0)^T$.

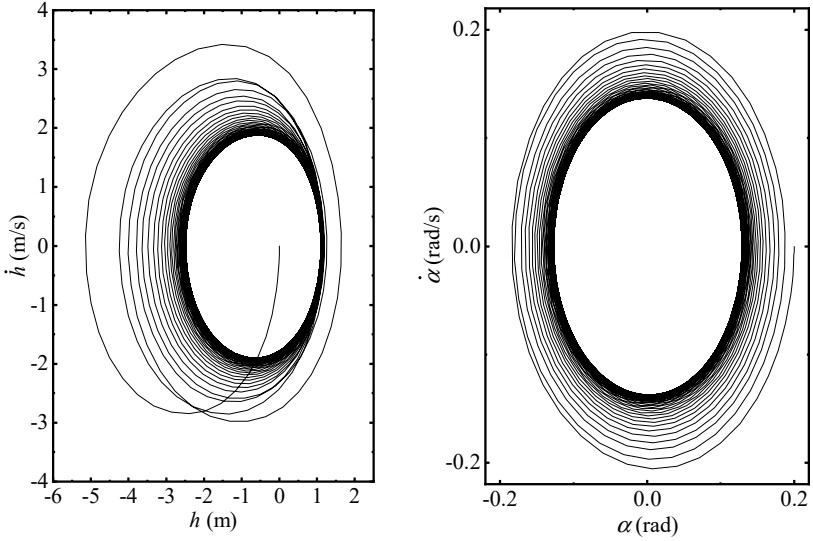

(**a**) Vibration LCO in vertical DOF   (**b**) Vibration LCO in torsional DOF

**Figure 9.** Phase trajectory diagram of LCO in vertical and torsional DOF at wind speed of 75 m/s and initial state of motion $\left(h, \dot{h}, \alpha, \dot{\alpha}\right)_2^T = (0, 0, 0.2, 0)^T$.

## 4. Analysis of Nonlinear Bifurcation and Limit Cycle Oscillation

Using the Runge–Kutta method discussed in the previous section, the LCO amplitude at a specific wind speed can be obtained through integration. Therefore, by selecting wind speed values within a certain range and substituting them into the reference case of the nonlinear motion system, we can integrate them to obtain a set of correspondences between wind speed and the stable LCO amplitude. This data can be used to plot the bifurcation curve of the flutter, as shown in Figure 10, which illustrates the relationship between wind speed and the stable LCO amplitude of the torsional degree of freedom for this particular operating condition. In this context, the abscissa U signifies the actual wind speed, while the ordinate A denotes the maximum amplitude of the limit cycle for the torsional degree of freedom.

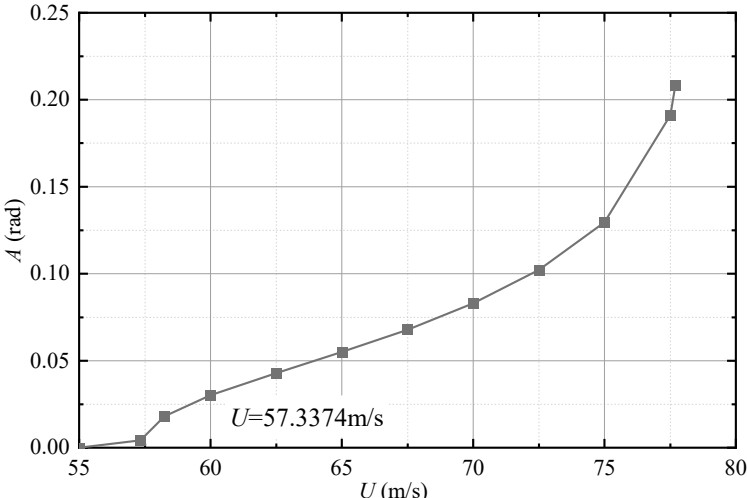

**Figure 10.** Bifurcation curves for the relationship between stable LCO amplitudes and wind speed.

At the flutter critical state $U_{cr} = 57.3374$ m/s, the nonlinear flutter system undergoes a Hopf bifurcation, and a stable LCO bifurcation curve appears on the right side of the bifurcation point. The LCO amplitude rises with the increasing wind speed. From the graph, it can be observed that after the wind speed exceeds $U = 77.7$ m/s, the trend of the limit cycle amplitude shows a steep increase, and stable limit cycle vibration cannot be obtained at higher wind speeds. Therefore, for this operating condition of the nonlinear flutter system, vibration divergence occurs when the wind speed exceeds $U = 77.7$ m/s.

Further research can be conducted on the bifurcation curve in Figure 10, which may reveal the presence of unstable LCO within the range of wind speeds $U < 77.7$ m/s, with amplitudes larger than the stable limit cycle. This can be investigated by setting larger initial conditions and performing integration calculations.

- Wind Speed: Wind speed $U = 75$ m/s (beyond the critical state);

Initial state of motion: $\left(h, \dot{h}, \alpha, \dot{\alpha}\right)_3^T = (0, 0, 0.45, 0)^T$.

By selecting a wind speed of $U = 75$ m/s and initial conditions of $\left(h, \dot{h}, \alpha, \dot{\alpha}\right)_3^T = (0, 0, 0.45, 0)^T$, the time response of the flutter system vibration can be computed.

As shown in Figures 11 and 12, under the given initial conditions, the vibration amplitude of the system initially decays and eventually stabilizes into a LCO vibration response after a period of time. Moreover, this stable LCO vibration response is consistent with the stable limit cycle response shown in Figures 4 and 5. The phase portraits of these two figures, representing the limit cycles, are plotted in Figure 13.

- Wind Speed: Wind speed $U = 75$ m/s (beyond the critical state);

Initial state of motion: $\left(h, \dot{h}, \alpha, \dot{\alpha}\right)_4^T = (0, 0, 0.452, 0)^T$.

The selected initial conditions of motion are slightly greater than the previous integration initial values. The resulting time-domain response of the system is calculated and shown in Figures 14 and 15.

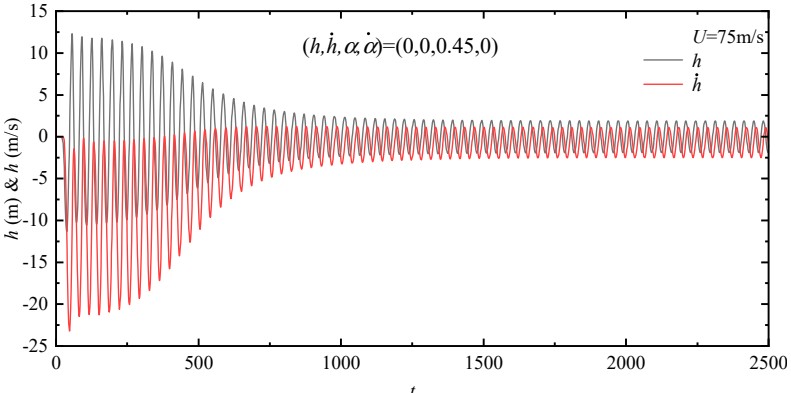

**Figure 11.** Continuous vibration of bridge flutter in vertical bending DOF at wind speed of 75 m/s and initial state of motion $\left(h, \dot{h}, \alpha, \dot{\alpha}\right)_3^T = (0, 0, 0.45, 0)^T$.

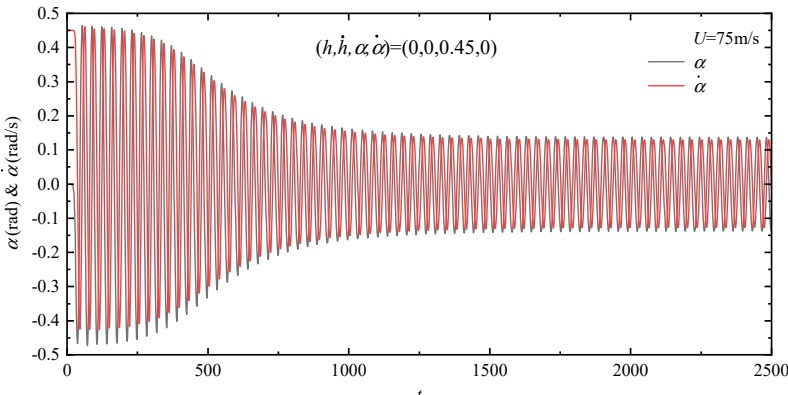

**Figure 12.** Continuous vibration of bridge flutter in torsional DOF at wind speed of 75 m/s and initial state of motion $\left(h, \dot{h}, \alpha, \dot{\alpha}\right)_3^T = (0, 0, 0.45, 0)^T$.

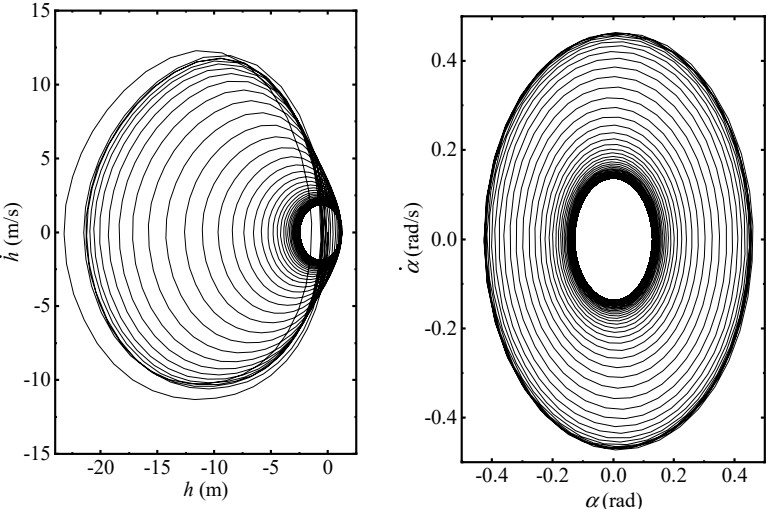

(**a**) Vibration LCO in vertical DOF     (**b**) Vibration LCO in torsional DOF

**Figure 13.** Phase trajectory diagram of LCO in vertical and torsional DOF at wind speed of 75 m/s and initial state of motion $\left(h, \dot{h}, \alpha, \dot{\alpha}\right)_3^T = (0, 0, 0.45, 0)^T$.

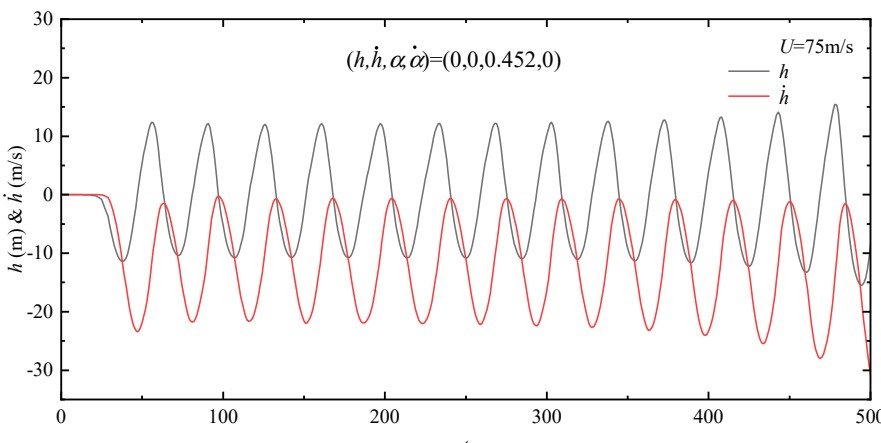

**Figure 14.** Continuous vibration of bridge flutter in vertical bending DOF at wind speed of 75 m/s and initial state of motion $\left(h, \dot{h}, \alpha, \dot{\alpha}\right)^{T}_{4} = (0, 0, 0.452, 0)^{T}$.

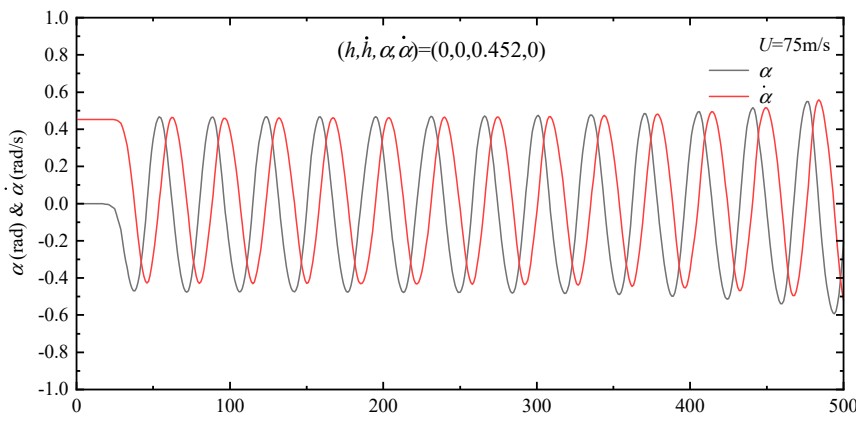

**Figure 15.** Continuous vibration of bridge flutter in torsional DOF at wind speed of 75 m/s and initial state of motion $\left(h, \dot{h}, \alpha, \dot{\alpha}\right)^{T}_{4} = (0, 0, 0.452, 0)^{T}$.

The vibration response of the system shows a slow increase in amplitude and max speed in both DOFs, indicating a gradual divergence of the oscillations. The vibration phase diagram is shown in Figure 16. A comparison with the motion phase diagram in Figure 13, which had the initial conditions $\left(h, \dot{h}, \alpha, \dot{\alpha}\right)^{T}_{3} = (0, 0, 0.45, 0)^{T}$ under the same wind speed, reveals the presence of an unstable limit cycle oscillation within the range of torsional amplitudes between 0.45 and 0.452. The vibration phase trajectories on either side of this LCO will gradually move away from it.

From the analysis above, at a wind speed of $U = 75$ m/s, the origin point of this nonlinear flutter system is an unstable focus. Additionally, there are two LCOs in the phase diagram. The smaller-amplitude LCO is stable, and vibration starting from the unstable focus will eventually converge to this stable LCO. The larger amplitude limit cycle is unstable one. Vibrations within the range between the unstable and stable LCOs will gradually approach the stable one, while vibrations with amplitudes greater than the unstable LCO will diverge.

In summary, the Runge–Kutta integration method can be used to solve flutter systems with nonlinear aerodynamic self-excitation and obtain bifurcating LCOs beyond the critical state. The stable LCO amplitude at a specific wind speed can be obtained through the steady-state time response, while the unstable LCO requires the following calculation steps: select a wind speed $U$ and an initial condition $(0, 0, \alpha_0, 0)^{T}$, calculate the time response of the nonlinear system vibrations, and continuously change the initial amplitude $\alpha_0$. When

the response above this value converges to the stable limit cycle and the response above this value diverges, it can be concluded that there exists an unstable limit cycle at the wind speed $U$, with an amplitude of $\alpha_0$. By substituting the parameter values of the reference case into the flutter Equations (4) and (5), the stable and unstable limit cycles corresponding to the wind speed can be obtained. Figure 17 shows the relationship between the limit cycle amplitudes and wind speed.

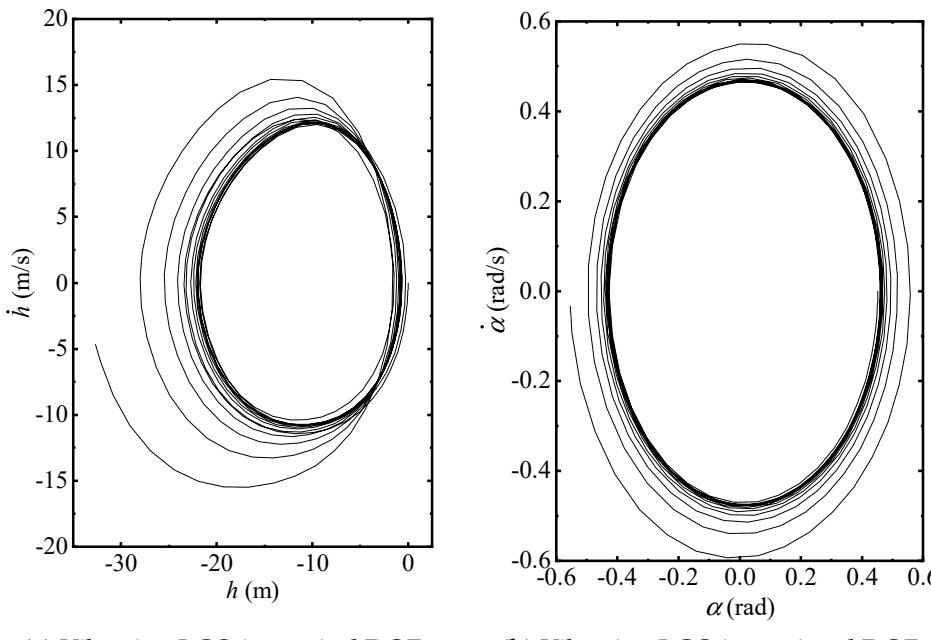

(**a**) Vibration LCO in vertical DOF  (**b**) Vibration LCO in torsional DOF

**Figure 16.** Phase trajectory diagram of LCO in vertical and torsional DOF at wind speed of 75 m/s and initial state of motion $\left(h, \dot{h}, \alpha, \dot{\alpha}\right)_4^T = (0, 0, 0.452, 0)^T$.

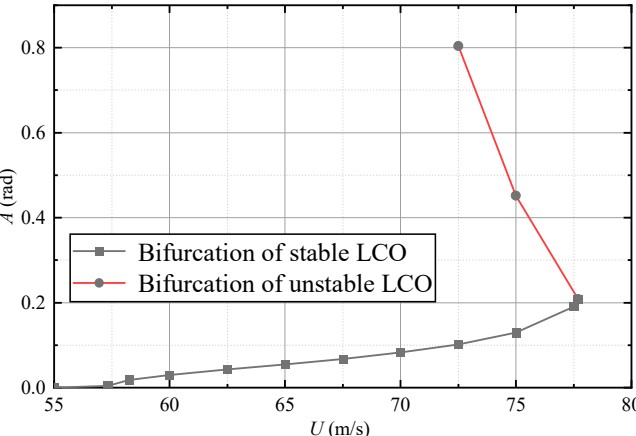

**Figure 17.** Bifurcation curves for the relationship between LCO amplitudes and wind speed.

Taking into account the parameter settings for the other three operating conditions in Table 2, the relationship between the LCO amplitude and wind speed after the supercritical Hopf bifurcation is calculated using the aforementioned method. The results are plotted in Figures 18 and 19.

As shown in the results of the LCO bifurcation curves, the flutter systems of all four operating conditions undergo supercritical Hopf bifurcation at the same critical value, leading to stable limit cycle oscillations on the side above the critical value. The stable LCO amplitude rises with the increasing wind speed. Comparing the curves of the reference case and case A, the two curves nearly overlap, but the stable LCO amplitude of case

A is relatively smaller than that of the reference case, and the unstable LCO amplitude is larger than that of the reference case. This indicates that applying stronger nonlinear damping on the torsional degree of freedom has a limited dissipating effect on limit cycle oscillations under supercritical flutter. Furthermore, comparing the curves of case A and case B, it can be observed that to achieve the same LCO amplitude, case B needs higher wind speeds. This suggests that considering the nonlinear stiffness of the bridge structure reduces the stable LCO amplitude and increases the unstable LCO amplitude under supercritical flutter, indicating a higher stability of the bridge compared to a system without considering structural nonlinear stiffness. Finally, in case C, by adding strong nonlinear damping on the vertical bending degree of freedom, the corresponding wind speed for the same LCO amplitude is the highest, demonstrating strong energy dissipation and suppression capabilities for supercritical flutter. In conclusion, by considering various nonlinear factors, the variation relationship between the limit cycle amplitude and wind speed for the four operating conditions has been analyzed. Among them, considering structural nonlinear stiffness and cubic damping on the vertical bending degree has the most significant influence on the LCO amplitude under supercritical flutter.

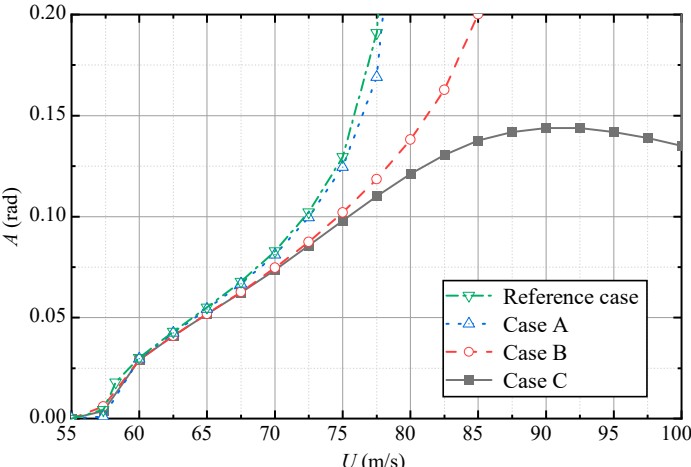

**Figure 18.** The Hopf bifurcation results of the calculation examples in Table 2, the relationship between the amplitude of the LCO and the wind speed.

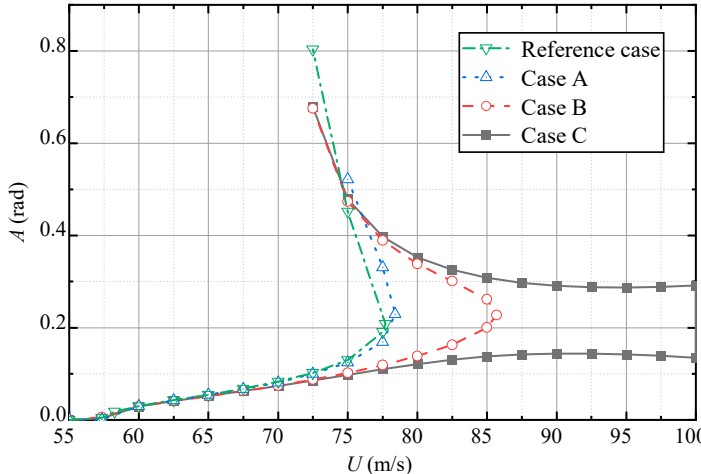

**Figure 19.** The Hopf bifurcation results of the calculation examples in Table 2, the relationship between the amplitude of the LCO and the wind speed, including stable and unstable LCO.

## 5. Conclusions

This paper considers the effects of nonlinear damping and structural stiffness on the limit cycle response of a long-span suspension bridge's flutter systems under supercritical

conditions, focusing on analyzing the influence of strong nonlinear damping and cubic stiffness on limit cycle oscillation at the flutter critical state. The nonlinear flutter self-excited force model proposed by Gao is considered, and the flutter derivatives obtained from wind tunnel tests are fitted using the least squares method to establish the nonlinear flutter self-excited force equation.

The flutter equation is transformed into state-space equations, and the flutter critical state is solved by analyzing the changes in matrix eigenvalues and obtaining the critical wind speed value. By using the Jacobian matrix of the state-space equations, the changes in matrix eigenvalues at the critical point are analyzed, and the supercritical Hopf bifurcation of the flutter system is demonstrated through vibration responses. Furthermore, the results of the four cases in Table 2 show that the flutter critical values for the systems are the same, indicating that supercritical Hopf bifurcation occurs at the same critical value. Finally, through a comparative analysis of the four case studies, the relationship between the limit cycle amplitude and wind speed under supercritical conditions is analyzed, demonstrating the effective flutter suppression and energy dissipation effects of cubic damping and stiffness.

The above studies indicate that in terms of engineering practicality, the flutter occurring in long-span suspension bridges is predictable and regular, displaying LCOs with amplitudes increasing as wind speed rises. To mitigate the detrimental effects of flutter on the bridge, increasing vertical bending and torsional damping of the bridge is an effective approach. By employing tuned mass dampers to apply strong nonlinear damping in both vertical bending and torsion motion dimensions, the amplitude of flutter oscillations in the bridge can be significantly reduced, thereby minimizing further damage caused by the displacement and deformation of the bridge. With the advancement of deep learning and machine learning, our research will leverage artificial neural network algorithms to conduct investigations, facilitating the prediction and evaluation of the vibrational response of bridges under aerodynamic forces.

**Author Contributions:** Conceptualization, J.L. and F.W.; methodology, J.L. and F.W.; software, J.L.; validation, J.L. and Y.Y.; writing—original draft preparation, J.L.; writing—review and editing, J.L., F.W. and Y.Y.; visualization, J.L.; supervision, F.W.; project administration, F.W.; funding acquisition, F.W. All authors have read and agreed to the published version of the manuscript.

**Funding:** This research was supported by the Key Laboratory of Disaster Forecast and Control in Engineering (Jinan University), MOE of China.

**Informed Consent Statement:** Not applicable.

**Data Availability Statement:** The data presented in this study are available on request from the corresponding author.

**Conflicts of Interest:** The authors declare no conflict of interest.

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
