# Peer review of "Study of Nonlinear Aerodynamic Self-Excited Force in Flutter Bifurcation and Limit Cycle Oscillation of Long-Span Suspension Bridge"

_applsci, doi:10.3390/app131810272_

Round 1

Reviewer 1 Report

Comments and suggestions are in the file attached.

Moderate editing of English language required.

Author Response

Dear Reviewer, Thanks for taking the time to review this manuscript. I really appreciate all your comments and suggestions!

Reviewer 2 Report

My comments on ApplSci-2511420:

The authors in this paper investigate the effects of nonlinear damping and structural stiffness on the limit cycle response of long-span suspension bridge flutter systems under supercritical conditions, focusing on analyzing the influence of strong nonlinear damping and cubic stiffness on the limit cycle oscillation in the critical state of the flutter. The nonlinear self-excited force model of the flutter proposed by Gao is considered, and the derivatives of the flutter obtained from the wind tunnel tests are fitted via the least-squares methodology to establish the nonlinear self-excited force equation of the flutter. Subsequently, the flutter equation is converted into state space equations, and the critical state of the flutter is appropriately unlocked by analyzing the changes in the eigenvalues of the matrix, and the critical value of the wind speed is suitably obtained and discussed. Using the Jacobian matrix of the state space equations, the changes in the eigenvalues of the matrix at the critical point are analyzed and the supercritical Hopf bifurcation of the flutter system is demonstrated through the vibration responses. Finally, through a comparative analysis of four case studies, the relationship between limit cycle amplitude and wind speed in supercritical conditions is analyzed and discussed. 

It is a fairly well-written paper and can be considered for possible publication in the special issue of “Wind Resistance of Long-Span Bridges and High-Rise Buildings” after the following raised issues are appropriately responded by the authors:

Major comments:

  1. As I see, Equations (1) and (2) display the translational and torsional motions of the bridge deck section based on the rigid section NOT the flexible one. Thereby, the main important question is whether the usage of a rigid-based model has sufficient accuracy for practical applications or whether it may lead to reasonable results for real structures. Please clarify based on the previously conducted works or other appropriate pieces of evidence. 

2.     According to Equations (1) and (2), the main reason for coupling between these two equations is the given drag forces (i.e., right-hand sides lateral force and torsional moment), while these two crucial terms were not cited. It implies that if these two terms are not directly contributed by the authors, they should be appropriately cited; otherwise, their bases should be explained more with more details.

3.     From Equation (14) to the last relation of the article, if a vector or matrix has been used in the formulations, this should be given in the non-italic bold format. Further, commonly, lowercase and uppercase letters are used for vectors and matrices; therefore, Fnonlinear should be modified to Fnonlinear (where Fnonlinear is a matrix-based parameter) for the sake of more consistency with the existing literature. And x should be modified to x since it is a vector. So please check the entire manuscript against this deficiency and make the required modifications. 

4.     No parts of the article have been compared with those of other works. Could the authors verify some parts of the calculations with those of other works to ensure the carried-out calculations and the utilized factors? 

5.     In Figures 17-19, it seems that the given y-label factor (i.e., A) has been not defined. If this is the case, please recheck and define that appropriately. 

6.     What are the main assumptions in constructing/using the given equations of motion (Equations 1 & 2)? Please list them carefully in the corresponding section where they have been introduced/presented for the first time. 

7.     The stability of the used numerical scheme (Runge-Kutta integration scheme) has been not displayed/discussed. How the authors can ensure the accuracy of the obtained results based on this numerical scheme? 

8.     No practical approach/way is suggested to enhance the stability of the structure against the wind loading. Can we increase the flutter stability of the wind through specific considerations (cross-sectional design, attached parts, and so on)? Please also clarify all the factors on this crucial parameter and how we can engineer that appropriately leading to a higher flutter velocity. 

9.     Three cases (Case A, Case B, and Case C) are considered according to the damping and stiffness factors given in Table 2 (the factors of the linear and nonlinear terms). In addition, the plotted results of the amplitude of dynamic angle (LCO) in terms of the wind velocity are provided in Figures 18 and 19; however, no discussion is given on which one is closer to the near-to-exact dynamic response of the real structure. Please clarify and display that in some detail. 

10.  The English of the paper is good; however, there are some grammatical errors and long sentences that require further attention from the authors. 

Minor comments:

  1. On page 1, the corresponding reference number(s) of “Scanlan et al.” should be given just after that. Please check and make the required modifications. 
  2. On page 2, the corresponding reference number(s) of “Gao et al.” should be given just after that. Therefore, it is revised to “Gao et al. [15]”. The same story also holds true for both “Zhang et al.” and “Wu et al.”. Please recheck them carefully and make the required modifications as well. 
  3. On page 16, the expression “Figure 18 and Figure 19” should be modified to “Figures 18 and 19”. 
  4. On page 16, the given x-label and y-label factors (i.e., “U” and “A”) should be presented in the italic manner since both of them are scalar. 

Please see the comment given above.

Author Response

(The authors gave the same response as above.)

Reviewer 3 Report

This is interesting and timely work. Some comments:

1.       The abstract is very easy to follow.

2.       The motivation of this study is clear.

3.       Introduction, Lines 51-77: the authors only list the methodologies in previous studies, how about their drawbacks and limitations?

4.       Introduction: ‘Based on the current research status, it is evident that nonlinear aerodynamic self-excitation exists, and the discrete values of the nonlinear components of flutter derivatives can be obtained through section model wind tunnel tests.’ This is existing efforts, what are the contributions in study?

5.       According to my knowledge, the methods and models applied are well developed, what are the key innovations?

6.       The overall analysis is reliable.

7.       What is the future perspective?

Not aware.

Author Response

(The authors gave the same response as above.)

Round 2

Reviewer 2 Report

I carefully read the responses of the authors to the former comments, as well as the modifications applied to the previous version of the paper. Most of them were explained and responded to reasonably; however, some others require further numerical work and reasonable explanations:

Issue#1: Previous query#7

In this query, I have asked the authors to explain how they ensure the stability of the used numerical scheme (Runge-Kutta integration scheme); however, they only explain that the fourth-order Runge-Kutta method meets the numerical simulations/predictions demands. I feel the given explanations by the authors alone are NOT satisfactory since they did NOT answer how the choice of the factor “h” and the number of time interval subdivisions could assure us that this method is numerically stable both conditionally and unconditionally. To this end, the authors are referred to the existing literature on the numerical stability of the Rung-Kutta approach.

Issue#2:

The authors have done their best to improve the English of the paper; however, there exist some style errors, grammatical erroneous, poor sentences, and typos that should be revised more. For guidance of the authors, some of them are given in the following:

-       On page 3, Equations (1) and (2), the presented brackets in these relations should be changed to parentheses for more consistency with the present literature.

-       On line 128, “Where” à “where”. Please note that the given “where” used after each relation should be started with a lowercase letter for the sake of more consistency with the present literature and published papers.

-       On line 140, “Where” à “where” (Based on the above-mentioned rule).

-       In Table 1 on page 4, the given units in the last column should be typed in a non-italic manner, preferably inside parenthesis. Hence, please modify them accordingly.

-       On line 170, the forces’ expressions “F(L)” and “F(M)” should be presented in an italic manner, except the given parentheses; therefore, we will arrive at “F(L)” and “F(M)”.

-       Through the paper manuscript, a free space should be exerted between the first bracket associated with the reference number and its follower word. For example, on line 117, the expression “inertia[25]” should be modified to “inertia [25]”; as another example, on line 124, the expression “cross-section[12,25]” should be changed to “cross-section [12,25]”. Therefore, the whole manuscript should be carefully checked against this style-type deficiency.

-       On line 69, “around bluff body” à “around the bluff body” (missing article “the”)

-       On lines 69 and 70, “wind tunnel test and numerical simulations” à “wind tunnel tests and numerical simulations” (for fixing the agreement mistake)

-       On line 48, the given reference style citation suffers from a slight mistake; actually, a dot symbol (i.e., “.”) has been missed after the appeared surnames of the authors. Therefore, “Scanlan et al [12]” should be modified to “Scanlan et al. [12]”. Please check the whole paper against this style-type typos.

-       On lines 36 and 37, the statement “known as limit cycle oscillations, in bridges has been indicated…” should be changed to “known as limit cycle oscillations in bridges, has been indicated…”.

-       On line 457, “through comparative analysis” should be changed to “through a comparative analysis”.

Please note that these are just samples from many errors that exist in the body of the paper and thereby the authors are highly encouraged to work more on the English text of the paper’s manuscript and carefully remove all deficiencies from the entire text of the paper.

Issue#3:

The advancements in machine learning and deep learning are unlocking new possibilities in the field of engineering sciences. These approaches are proving to be extremely useful in analyzing complex systems and estimating near-to-exact responses of various structures. In fact, the authors should consider discussing this as a hot topic for future works in their paper. They can cite various engineering applications to support their arguments, such as structural mechanics and mechanics, by using the following reference works:

(1)  Gasparin, A., Lukovic, S., Alippi, C.: Deep learning for time series forecasting: the electric load case. Fan B, Zhang Y, Chen Y, Meng L. Intelligent vehicle lateral control based on radial basis function neural network sliding mode controller.

(2)  Zhang, G.Q., Wang, B., Li, J. and Xu, Y.L. The application of deep learning in bridge health monitoring: A literature review.

(3)  Wu Z, Luo G, Yang Z, Guo Y, Li K, Xue Y. A comprehensive review on deep learning approaches in wind forecasting applications.

(4)  Mallikarjuna SB, Shivakumara P, Khare V, Basavanna M, Pal U, Poornima B. Multi‐gradient‐direction based deep learning model for arecanut disease identification.

(5)  Zhang R, Meng L, Mao Z, Sun H., Spatiotemporal deep learning for bridge response forecasting.

Issue#4:

In addition, the application of “artificial intelligence analysis” in developing appropriate codes, algorithms, and solutions for engineering problems is going to be increasingly intensified day by day. By utilizing AI, engineers can effectively analyze data gathered from simulations, sensor networks, and other sources to enhance the design and performance of structures. With AI's assistance, engineers can accurately predict the behavior of structures under different loads and conditions, allowing them to make informed decisions regarding design and construction. This invaluable tool empowers engineers to create structures that are not only innovative and efficient but also safe and reliable. By this virtue, the authors are advised to elucidate the above-mentioned explanations through referring to the following relevant works:

(1)  Hsiao, I. H., & Chung, C. Y. AI-infused semantic model to enrich and expand programming question generation.

(2)  Zhang Y, Yuen KV. Review of artificial intelligence-based bridge damage detection.

(3)  Jia, Z., Wang, W., Zhang, J., & Li, H. Contact High-Temperature Strain Automatic Calibration and Precision Compensation Research.

(4)  Wang, M., Yi, H., Jiang, F., Lin, L., & Gao, M. Review on offloading of vehicle edge computing.

(5)  Shafqat, S., Majeed, H., Javaid, Q., & Ahmad, H.F. Standard ner tagging scheme for big data healthcare analytics built on unified medical corpora.

Issue#5:

-       In line 229, the authors are referring to a specific approach (i.e., “Theodoron method”), however, no reference number has been provided for that. For beginner readers, providing some appropriate reference works for this method will be helpful to follow the contents of the article more easily.

-       On line 113, at least an appropriate reference should be given for “Scanlan's proposed model”. If the reference number of this is 25, it should be translocated from the end of the paragraph to just after the aforementioned expression.

Issue#6: Previous query#1

In fact, I was NOT satisfied with the given explanations of the authors for the former query#1. Let us ask my question in another way. As I see, the bridge deck section presented in Figure 1 is modeled by a 2DOFs-rigid-body. However, we know this is surely a flexible structure, and concerning the flutter instability phenomenon, a combination of flexural and torsional vibrations and possibly the longitudinal vibration plays a crucial role in the dynamic instability of the bridge deck. Why the authors have not considered a flexible deck for analysis of the problem? Only citing a given reference work would not be satisfactory. More precisely, I am looking for what we missed, both structurally and mechanically, through transforming from a flexible deck to a rigid one and how considering the flexible deck (i.e., more accurate equations of motion of both transverse and torsional vibrations of the deck based on beam theories). Actually, we are reducing an elastic system of infinite number of DOFs to a rigid one with only 2DOFs, which requires some major assumptions that we want to hear from the authors. I hope this will help you to further think about this more carefully and could thoughtfully display to us. In addition, the following works pertinent to coupled torsional-transverse vibrations as well as transverse vibrations of pre-twisted beams can be referenced in your explanations on the reduction of the model:

(1)  Vlase S, Marin M, Scutaru ML, Munteanu R. Coupled transverse and torsional vibrations in a mechanical system with two identical beams. AIP Advances. 2017 Jun 1;7(6).

(2)  Mu B, Kiani K. Surface and shear effects on spatial buckling of initially twisted nanowires. Engineering Analysis with Boundary Elements. 2022 Oct 1;143:207-18.

(3)  Ma X, Kiani K. Spatially nonlocal instability modeling of torsionaly loaded nanobeams. Engineering Analysis with Boundary Elements. 2023 Sep 1;154:29-46.

(4)  Li M, Wang C, Kiani K. Spatial vibrations and instability of axially loaded–torqued beam-like nanostructures via surface elasticity theory. Engineering Analysis with Boundary Elements. 2023 Apr 1;149:1-7.

(5)  Banerjee JR, Su H. Free transverse and lateral vibration of beams with torsional coupling. Journal of Aerospace Engineering. 2006 Jan;19(1):13-20.

(6)  Tian Y, Zhang X, Chen C, Fang B, Cao D. Vibration isolation performance of a beam clamped with torsional quasi-zero-stiffness isolator. Journal of Vibration and Control. 2023 Apr 28:10775463231173169.

Issue#7:

In order to suitably address comment#6, it is important to display all the assumptions made in constructing Equations (1) and (2). These assumptions can be appropriately itemized in a paragraph provided before or after Equations (1) and (2). By doing so, readers will have a better understanding of the underlying assumptions and limitations of the model.

The authors have done their best to improve the English of the paper; however, there exist some style errors, grammatical erroneous, poor sentences, and typos that should be revised more. For guidance of the authors, some of them are given in the following:

-       On page 3, Equations (1) and (2), the presented brackets in these relations should be changed to parentheses for more consistency with the present literature.

-       On line 128, “Where” à “where”. Please note that the given “where” used after each relation should be started with a lowercase letter for the sake of more consistency with the present literature and published papers.

-       On line 140, “Where” à “where” (Based on the above-mentioned rule).

-       In Table 1 on page 4, the given units in the last column should be typed in a non-italic manner, preferably inside parenthesis. Hence, please modify them accordingly.

-       On line 170, the forces’ expressions “F(L)” and “F(M)” should be presented in an italic manner, except the given parentheses; therefore, we will arrive at “F(L)” and “F(M)”.

-       Through the paper manuscript, a free space should be exerted between the first bracket associated with the reference number and its follower word. For example, on line 117, the expression “inertia[25]” should be modified to “inertia [25]”; as another example, on line 124, the expression “cross-section[12,25]” should be changed to “cross-section [12,25]”. Therefore, the whole manuscript should be carefully checked against this style-type deficiency.

-       On line 69, “around bluff body” à “around the bluff body” (missing article “the”)

-       On lines 69 and 70, “wind tunnel test and numerical simulations” à “wind tunnel tests and numerical simulations” (for fixing the agreement mistake)

-       On line 48, the given reference style citation suffers from a slight mistake; actually, a dot symbol (i.e., “.”) has been missed after the appeared surnames of the authors. Therefore, “Scanlan et al [12]” should be modified to “Scanlan et al. [12]”. Please check the whole paper against this style-type typos.

-       On lines 36 and 37, the statement “known as limit cycle oscillations, in bridges has been indicated…” should be changed to “known as limit cycle oscillations in bridges, has been indicated…”.

-       On line 457, “through comparative analysis” should be changed to “through a comparative analysis”.

Please note that these are just samples from many errors that exist in the body of the paper and thereby the authors are highly encouraged to work more on the English text of the paper’s manuscript and carefully remove all deficiencies from the entire text of the paper.

Author Response

Dear Reviewer,

Thank you very much for taking the time to review this manuscript. I really appreciate all your comments and suggestions! Please find my itemized responses in the attachment and my revisions/corrections in the re-submitted files.

Sincerely.

Reviewer 3 Report

Thanks to the authors' hard work, almost all my comments were revised. No further review process is needed.

It looks good to me.

Author Response

Dear Reviewer,

Thank you very much for taking the time to review this manuscript. I really appreciate all your comments and suggestions!

Sincerely.

Round 3

Reviewer 2 Report

The paper can be accepted.